# Genetic Prediction of Erectile Dysfunction Caused by Statins Among Malaysian Cardiac Patients

**DOI:** 10.3390/ijms26178447

**Published:** 2025-08-30

**Authors:** Naam Bahjat Ahmed Adeeb, Hadeer Akram Al-Ani, Nur Aizati Athirah Daud, Ruzilawati Abu Bakar, Imran Ahmad, Dzul Azri Mohamed Noor, Zilfalil Bin Alwi

**Affiliations:** 1Human Genome Centre, School of Medical Sciences, Universiti Sains Malaysia, Kubang Kerian, Kota Bharu 16150, Kelantan, Malaysia; ruzila@usm.my (R.A.B.); profimran@usm.my (I.A.); zilfalil@usm.my (Z.B.A.); 2College of Medicine, Ibn Sina University of Medical and Pharmaceutical Sciences, Baghdad 79G3+3RR, Iraq; 3School of Medicine, University of California Davis, One Shields Avenue, Davis, CA 95616, USA; haalani@ucdavis.edu; 4School of Pharmaceutical Sciences, Universiti Sains Malaysia, Gelugor 11800, Pulau Pinang, Malaysia; dzulazri@usm.my

**Keywords:** erectile dysfunction, atorvastatin, *NR2F2-AS1*, *NOS3*, *CYP17A1*, *CYP19A1*

## Abstract

Erectile dysfunction (ED) is a significant disorder commonly found in patients with cardiovascular diseases and diabetes mellitus. Recent updates have indicated that statins may also contribute to an increased risk of ED. This cross-sectional genetic study involved 246 cardiac patients attending normal visits at Hospital Universiti Sains Malaysia (USM) Kubang Kerian outpatient clinics. The patients were categorized into four groups: statins with ED (group 1), statins without ED (group 2), diabetes with statin without ED (group 3), and healthy patients without diabetes mellitus and statin use (group 4). Six genes were hypothesized to influence ED: *CYP19A1*, *CYP17A1*, *SIM1*, *TP53*, *NR2F2*, and *NOS3*, with different polymorphisms and variants investigated in this study. Overall, statin therapy was found to have a negative impact on ED by affecting *NR2F2-AS1* and *NOS3*. However, atorvastatin showed varying effects on ED for all genes, with the highest impact observed with *CYP17A1* and the lowest with *CYP19A1*. In conclusion, this study revealed novel findings related to genetic factors influencing ED in Malaysian males during statin use.

## 1. Introduction

Erectile dysfunction (ED) is a common symptom of male sexual dysfunction and is a prevalent but inadequately treated condition [1]. Nordin et al. [2] reported that 81.5% of the Malaysian population experience ED, with varying degrees of severity: mild (17%), mild–moderate (23.8%), moderate (11.3%), and severe (29.5%). However, this prevalence has potentially been overestimated, because ED was determined based on self-reported data rather than clinical confirmation by a physician. Although statins are medicines associated with a high number of reported adverse reactions [3,4], no one has examined the impact of chronic use of such medicines on ED, or, more importantly, predicted genetic factors among Malaysian men. There is ongoing debate regarding the clinical effects of statins on male sexual function, with some studies suggesting that statins may decrease sexual function [5,6], have no effect [7], or even protect sexual function [8]. Another study on the modes of actions and drug interactions of statins has shown that they do influence ED [9].

Scholars have investigated the influence of genetic relationships on sexual dysfunction due to the inhibition of testosterone production among males. Genetic studies by Aguirre et al. [10] and Roosenboom et al. [11] have demonstrated a significant link between genetic factors and ED. Several genes have been linked to a potential reduction in testosterone levels and erectile function through hormonal, neuroendocrine, and vascular mechanisms. *CYP19A1* encodes the aromatase enzyme, which converts testosterone to estrogen, while increased aromatase activity disrupts the hormonal balance and indirectly reduces testosterone levels, affecting sexual function [10,12]. *CYP17A1* produces a key steroidogenic enzyme involved in converting cholesterol to androgen precursors, and reduced expression or inhibition directly lowers testosterone synthesis [13,14]. Meanwhile, the *SIM1* gene encodes a transcription factor that modulates hypothalamic pathways related to libido and sexual behavior, with dysregulation potentially impairing luteinizing hormone signaling and reducing testosterone [15]. *TP53* encodes the p53 protein, which affects Leydig cell function and testosterone production, thereby influencing sexual performance [16], while *NR2F2* encodes COUP-TFII, a nuclear receptor involved in androgen receptor signaling and testicular development. Its dysregulation may reduce testosterone responsiveness and vascular erectile function [17]. Lastly, *NOS3* encodes endothelial nitric oxide synthase (eNOS), responsible for nitric oxide production, which is essential for penile vasodilation. Reduced *eNOS* expression leads to poor nitric oxide availability and compromised erectile function [18].

Information on statins generally covers the control of cholesterol synthesis, where cholesterol is the main source of testosterone production. However, there is a lack of genetic studies directly investigating the genetic influence of statins and their types on erectile dysfunction (ED). The hypothesis of the adverse reaction of statins on the ED can also be justified within the gene–drug interactions. The present study investigated several genes mentioned earlier that were hypothesized to interact with statins. This study aimed to examine the impact of genetic variants in *CYP17A1*, *CYP19A1*, *SIM1*, *NOS3*, *NR2F2,* and *TP53* genes on the sexual function score among Malaysian male statin users.

## 2. Results

### 2.1. Demographic Characteristics

A total of 246 patients were included in the current study. The percentage of patients who smoke and consume alcohol is 20.3% and 1.6%, respectively. The majority of patients are Malays, accounting for 94.7% of the total, and the majority are also married (93.9%). Most patients have a college degree (55.7%) as their highest level of education, followed by secondary (41.1%) and primary (3.3%) school education. The means (±SD) for age, weight, height, and BMI are 50.58 (±8.204), 78.55 (±16.409), 1.66 (±0.09), and 28.329 (±5.528), respectively, for each group, while the medians for age, weight, height, and BMI are 50.58, 78.55, 1.66, and 28.329, respectively.

### 2.2. Statin Therapy and Dosing Regimens

The majority of patients in the study were receiving statin therapy, with a percentage of 76.4%. Atorvastatin was more commonly prescribed than simvastatin (52% vs. 22.8%), with the statin dose ranging from 10 mg to 80 mg, with 20 mg being the most common dose (38.6%), followed by 40 mg (33.3%). The mean (±SD) duration of statin therapy for 187 patients was 27.32 (±34.878) months, with a median of 13 months.

### 2.3. ED and Alleles

The average (±SD) International Index of Erectile Function (IIEF) score was 47.768 (±13.975), with a median of 49.50. Table 1 shows the distribution of SNP alleles among patients with and without ED. In patients with ED, a higher proportion had normal alleles, with the exception of SNP4_rs1042522 and SNP5_rs803580.

### 2.4. Association Analysis Between SNPs and IIEF Score of ED

The groups were analyzed to determine the impact of the variants on ED based on physicians’ diagnosis and IIEF scores (Table 2, Figure 1). In group 1, the presence of allele GT or TT for SNP5_rs803580 showed a significant effect on ED, particularly in statin users, leading to a 4.981-fold reduction in IIEF scores compared to those with the reference allele (TT). Similarly, the presence of allele TG or GG for SNP6_rs1799983 in group 1, especially in statin users, resulted in a 6.357-fold reduction in IIEF scores compared to those with the reference allele (GG). There was no significant effect observed for SNP1_rs4919686, SNP2_rs17703883, SNP3_rs57989773, and SNP4_rs1042522 on ED risk across all groups. The presence of SNP5_rs803580 and SNP6_rs1799983 increased the risk of ED, particularly in statin users, based on the study outcomes.

### 2.5. Association Between Statin Types and IIEF Scores

Table 3 demonstrates the relationship between the type of statin and group and the risk of ED. The results indicate that, in group 1, atorvastatin had a significant impact on the risk of ED, increasing it 10.566-fold compared to the control group. There was no significant effect observed for simvastatin on ED. These findings suggest that the type of statin used may have a significant effect on the occurrence of ED in the patients included in this study.

### 2.6. Interaction Effects of Statin Types and Marker SNPs on the IIEF Score

The findings indicated a notable impact on ED of the interaction between atorvastatin and specific genetic variants (SNP1_rs4919686 variants AC or CC, SNP2_rs17703883 variants TC or CC, SNP3_rs57989773 variants TC or CC, SNP4_rs1042522 variants CG or CC, SNP5_rs803580 variants TC or CC, and SNP6_rs1799983 variants TG or GG) in group 1 (statin users). This suggests that the presence of certain genetic variants, influenced by atorvastatin, may exacerbate ED by factors of −8.585, −4.611, −8.084, −5.271, −6.163, and −7.858 compared to the control group (Table 4).

## 3. Discussion

### 3.1. Prediction of Genetic Effects on ED

Genetic variations among patients play a role in sexual dysfunction, with genes related to steroidogenesis and sexual functions impacting erectile function in males [19]. While many studies have explored the influence of genetic variation on erectile function, few have examined the impact of medications, such as lipid-lowering agents like statins. Various genes, including *MTHFR* [20], *NOS1* [21], *CYP17A1* [19], *APOC3*, *APOB*, *LDLR*, and *LPL* [22], as well as *CLDN5*, *COL7A1*, *LDHA*, *MAP2K2*, *RETSAT*, *SEMA3A*, *TAGLN*, and *TBC1D1* [23], have been linked to erectile dysfunction. However, these genetic factors are not solely related to medication side effects and may vary among different populations and races.

One significant polymorphism examined in this study is rs1799983 of the *NOS3* gene, which has been associated with erectile dysfunction [24]. Perticarrara Ferezin et al. [21] found that the *NOS* gene influences the erectile function in cardiac patients, with significant effects observed in the clinical group compared to the control group. Another important polymorphism is rs803580 of the *NR2F2-AS1* gene, which is involved in sexual gonad development [25]. While Hattori and Fukami identified that variants of *NR2F2* are associated with testicular sex development and testosterone secretion, the impact of this gene on erectile dysfunction has not been extensively studied. Some studies have explored the influence of *NR2F2* on testis development and testosterone production [17], but with different variants than those examined in this study. Overall, the findings of this study align with previous research on the genetic factors influencing erectile dysfunction, particularly in relation to statin use.

### 3.2. Statin Types and ED

Several studies have explored the impact of statins on ED, with most focusing on animal models [13,26], limiting their applicability to humans. Statins reduce cholesterol synthesis by inhibiting HMG-CoA reductase, which in turn lowers the availability of cholesterol—a crucial precursor for testosterone biosynthesis. Consequently, statin therapy has been associated with reduced testosterone levels. Several human studies have reported significant decreases in serum testosterone following statin initiation, with reductions of 8.68 nmol/L and 5.78 nmol/L observed by Nakayama et al. (2021) and Hsieh and Huang (2016), respectively [27,28].

This study utilized genetic techniques to analyze human data and laboratory results to assess the association between statins and ED. The previous literature primarily consists of reviews and meta-analyses of ED but lacks investigations into genetic effects [29,30].

Some researchers have reported the positive effects of statins on erectile parameters [31], while others have found varying influences of different statin types on erectile function. This can be attributed to several reasons, such as pharmacodynamics, pharmacokinetics, and genetic differences [32]. The present study found a significant association between atorvastatin and ED across all genotypes and alleles studied. This aligns with previous research highlighting atorvastatin’s adverse effects on prostate function [33].

This study also considered genetic factors contributing to the effects of atorvastatin on ED, in contrast to previous studies. Genetic variations, particularly in the *NOS3* gene, were found to play a role in the development of ED. The NOS3 gene exhibited an apparent relationship with the development of vascular functions [34]. Moreover, this gene influences sexual performance and testosterone levels, especially among older subjects [35]. Perticarrara Ferezin et al. [21] revealed the significant effects of polymorphisms of the NOS gene on the IIEF score among males in Brazil. The association found in this study, however, is justified by the small sample size of patients with gene variants. There is a need for studies to investigate the effect of these genes with atorvastatin in a prospective design to derive more robust associations and mechanisms on the ED.

Additionally, genetic variants in genes including *SIM1*, *TP53*, and *NR2F2* are associated with ED, but their interaction with atorvastatin use has not been extensively studied. Jorgenson et al. [15] highlighted the influence of genetic variants, including the SIM1 gene (using several alleles), on ED among patients in the United Kingdom, showing that SIM1 rs57989773 variation impaired sexual drive and the central regulation of testosterone [15]. However, the present study identified no significant impact of statins on the ED for this SIM1. The TP53 rs1042522 variant may reduce Leydig cell survival, and under statin-induced cellular stress, this could further limit testosterone secretion [36,37]. NR2F2-AS1 rs803580 exhibited a significant association with ED in statin users. This variation influences the expression of COUP-TFII, which is important for testicular function and androgen signaling. The presence of this variation together with statin use may further potentiate the testosterone-lowering effect of statins [17,25]. This result, however, needs to be interpreted with caution due to modest effect sizes, and the study may need to be replicated in larger cohorts.

With regards to pharmacogenetic predictors, Su et al. identified the genetic basis of the lipid-lowering agent, which was significantly associated with ED [22]. They also found that the variants of the genes influenced ED and associated illnesses (like prostate hyperplasia, prostate cancer, infertility, and others). They reported the causality, but there is a need for more information about the mechanism of action between statins and the genes that induce ED. Inversely, our study selected the genes most related to ED due to cholesterol synthesis, sex hormone production, and gonad development. Furthermore, this study determined the associations between statins and gene variants of the ED within drug–gene interactions.

Simvastatin exhibited no significant impact on ED among the study groups. This is because the variations in the modes of actions and susceptible genes are comparable to those of atorvastatin, due to its physicochemical and pharmacological properties. Simvastatin is a prodrug: it must be activated in the liver to become pharmacologically active, which introduces variability in the responses and enhances interactions with hepatic gene expression [13]. Atorvastatin is already active when administered; it does not require metabolic conversion, resulting in more predictable pharmacokinetics and less reliance on hepatic enzymes [3,5].

The main limitations of this study are as follows: the cross-sectional study design, which minimizes the cause and effect of ED linked to statins; the use of laboratory measurements, especially of testosterone and other biomarkers; the lack of a pre–post evaluation; the small sample size of the groups, especially in the statin and ED groups, which means it was not possible to address key confounding factors such as age, comorbidities, and weight; the selection of a dummy method in the linear regression, which leads to modest effect sizes; and the limitation regarding the selection of different races and geographic distribution, which minimized the generalization of the study results. Most of the participants in this study were Malays; therefore, the influence of ethnicity is a significant generalization limitation. The authors of this study recommend involving more ethnicities in a larger cohort to determine the differences in the incidence and genetic associations with ED.

## 4. Materials and Methods

### 4.1. Study Design

This study is a cross-sectional genetic association study, supported by similar studies from the literature [2,4,38,39]. Patients were selected based on regular clinical appointments and met the inclusion and exclusion criteria. Blood serum samples were collected with patient approval and nurse assistance. All participating patients signed consent forms. Physicians confirmed diagnoses, particularly for those with ED, and a self-administered questionnaire was used to gather information on patients’ eligibility, demographics, and medical/medication history.

This study took place at the Family Medicine Clinic of Hospital Pakar Universiti Sains Malaysia (HPUSM) Kubang Kerian. The primary target population included male patients aged 18 years and older attending for regular follow-ups who were on any dose of statin medication. Patients were excluded from this study if they declined to give consent or were diagnosed with prostate diseases, cancers, or other sexual disorders. The patients were divided into four groups: (1) patients taking statins and experiencing erectile dysfunction (ED); (2) patients taking statins and not experiencing ED; (3) patients taking statins who were diagnosed with diabetes mellitus but not ED; and (4) individuals not using statins and or experiencing ED.

### 4.2. Data Collection

Patient information included demographic characteristics, medical history, concurrent diseases, and medication history. To assess the severity of ED, all patients completed the IIEF questionnaire [40]. The questionnaire was translated from English to Malaysian by qualified translators to ensure a better understanding and accurate results. The erectile function component of the IIEF questionnaire was the focus of this study, as it is directly related to the effects of statins. This component consists of six questions related to erectile function, rated on a Likert scale: (0) no sexual activity; (1) rarely or never; (2) a few times (less than half the time); (3) sometimes (about half the time); (4) most of the time (more than half the time); and (5) almost always or continuously. Erectile dysfunction was assessed based on the IIEF score, with lower scores indicating poorer erectile function. Informed consent was obtained from all patients prior to recruitment, and blood samples were collected by trained personnel, with 5 mL of blood drawn from each patient.

### 4.3. DNA Extraction

DNA extraction was carried out using QIAamp DNA blood kits (QIAGEN, Victoria, Australia) following the instruction manual and employing the restriction fragment length polymorphism (RFLP) technique. The incubation protocols for each restriction enzyme varied, as shown in Table 5 below.

Procedure: Agarose gel was placed in an electrophoresis tank with 1X TBE buffer. The first well was loaded with a 100 bp DNA ladder mixture consisting of 1 μL of 100 bp DNA ladder, 1 ul of 6X loading dye, 1 ul of 100x SYBR green, and 3 ul of 1x TBE buffer. Each sample was prepared by mixing 1 ul of 6X loading dye, 1 ul of 100x SYBR green, and 4 ul of PCR amplicons or digested PCR amplicons, and then loaded into separate wells. The gel was electrophoresed for 40 min at 100V and 400 mA. The agarose gel was visualized under UV light using a UV transilluminator (Wealtec Corp., Sparks, NV, USA), and images were captured using an Alpha Digi Doc Tm system. Collected data was stored in a separate database for easy and efficient searching.

### 4.4. DNA Sequencing

DNA sequencing was performed on the PCR product at Apical Scientific Sdn Bhd (Selangor, Malaysia), to verify the genotypes.

### 4.5. Selection of Genotypes and Primer Design

Genotypes were chosen from literature studies on erectile dysfunction (ED) and disruptions in testosterone hormone levels. The *CYP19A1* and *CYP17A1* enzymes play a crucial role in synthesizing sex hormones, which are important for ED development. The *SIM1*, *TP53*, *NR2F2,* and *NOS3* genes were also examined due to their involvement in sexual function.

Bioinformatic tools are crucial for identifying suitable primers for specific genomic regions containing relevant polymorphisms. The *CYP17A1*, *CYP19A1*, *NOS3*, *TP53*, *NR2F2-AS1*, and *SIM1* genes, along with their corresponding SNP polymorphisms (rs4919686, rs17703883, rs1799983, rs1042522, rs8023580, and rs57989773), were analyzed. Blast software was primarily utilized in the bioinformatic tool to align nucleotides and design primers tailored to homozygous, heterozygous, and codominant alleles. Designing primers for genotyping polymorphisms in specific genes is a crucial aspect of the process, involving the identification of polymorphisms within gene sequences using bioinformatics methods. This ensures accurate amplification during PCR by targeting specific regions. Primers selection is influenced by the type of genetic variation, allele frequencies, and genotyping technique. Computational analysis was used to confirm primer specificity to prevent amplification of unintended genomic regions.

### 4.6. Data Analysis

SPSS version 24 was used to analyze all collected data. Patient characteristics were presented using descriptive statistics in terms of frequency and percentages. The IEEF score was used to assess ED, with a physician evaluating and assigning a score for each patient based on an IIEF evaluation. Linear regression analysis (dummy method) with a 95% CI was conducted to determine the association between genotypes of *CYP17A1*, *CYP19A1*, *NOS3*, *TP53*, *NR2F2-AS1*, and *SIM1* and ED, as well as the type of statin used. Results with a p-value less than 0.05 were considered statistically significant.

## 5. Conclusions

In conclusion, this study showed that there is a significant association between genetic factors and ED in Malaysian males using statins. In this study, atorvastatin showed a potential risk of adverse effects related to erectile function as compared to simvastatin, especially in the presence of genetic variations, with the effect more pronounced in the CYP17A1 gene compared to the CYP19A1 gene. The authors of this study support further investigations into the genetic variations associated with statins and ED in a larger sample size and across multiple populations in Malaysia and other countries. In addition, a time-dependent study design (such as longitudinal studies or trials) and advanced pharmacogenomic screening and genotyping techniques are recommended.

## Figures and Tables

**Figure 1 ijms-26-08447-f001:**
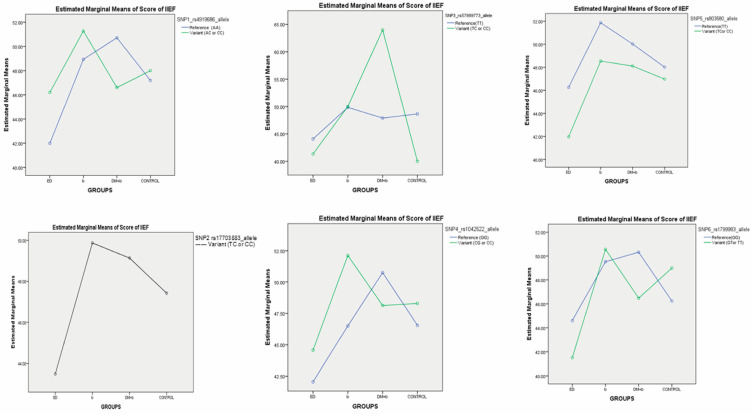
Comparison of IIEF mean scores between subjects with variant and reference alleles for all subject groups.

**Table 1 ijms-26-08447-t001:** Alleles and SNPs.

SNPs			Erectile Dysfunction	
		Total (N = 246, %)	NO = 186 (%)	YES = 60 (%)
*CYP17A1*: rs4919686 genotype	AA	221 (89.8)	166 (89.2)	55 (91.7)
	AC	13 (5.3)	10 (5.4)	3 (5)
	CC	12 (4.9)	10 (5.4)	2 (3.3)
*CYP19A1*: rs17703883 genotype	TC	224 (91.1)	170 (91.4)	54 (90)
	CC	22 (8.9)	16 (8.6)	6 (10)
*SIM1*: rs57989773 genotype	TT	237 (96.3)	181 (97.3)	56 (93.3)
	TC	9 (3.7)	5 (2.7)	4 (6.7)
*TP53*: rs1042522 genotype	GG	57 (23.2)	44 (23.7)	13 (21.7)
	CG	127 (51.6)	97 (52.2)	30 (50)
	CC	62 (25.2)	45 (24.2)	17 (28.3)
*NR2F2-AS1*: rs803580 genotype	TT	96 (39)	76 (40.9)	20 (33.3)
	TC	128 (52)	91 (48.9)	37 (61.7)
	CC	22 (9)	19 (10.2)	3 (5)
*NOS3*: rs1799983 genotype	GG	174 (70.7)	129 (69.4)	45 (75)
	GT	65 (26.4)	51 (27.4)	14 (23.3)
	TT	7 (2.8)	6 (3.2)	1 (1.7)
**Total**		**246**	**186**	**60**

**Table 2 ijms-26-08447-t002:** Variant and group interaction effects on ED.

Variants	Groups	Unstandardized Coefficients		Standardized Coefficients	t	Sig.	95.0% CI for B	
		B	Std. Error	Beta			Lower Bound	Upper Bound
SNP1 rs4919686 *	(Constant)	48.284	3.208		15.051	0.000	41.965	54.604
	Group 1	−6.037	3.195	−0.207	−1.890	0.060	−12.33	0.256
	Group 2	2.410	2.964	0.094	0.813	0.417	−3.429	8.249
	Group 3	0.555	3.055	0.020	0.182	0.856	−5.462	6.572
	Group 4	0.174	3.280	0.006	0.053	0.958	−6.286	6.635
SNP2 rs17703883 **	(Constant)	41.606	9.141		4.551	0.000	23.599	59.613
	Group 1	−0.301	4.334	−0.019	−0.070	0.945	−8.839	8.236
	Group 2	4.940	4.638	0.325	1.065	0.288	−4.195	14.076
	Group 3	3.730	4.653	0.230	0.802	0.424	−5.436	12.896
	Group 4	3.456	4.662	0.205	0.741	0.459	−5.728	12.640
SNP3 rs57989773 ***	(Constant)	46.830	4.631		10.113	0.000	37.708	55.952
	Group 1	−5.270	4.406	−0.177	−1.196	0.233	−13.950	3.410
	Group 2	4.387	4.652	0.151	0.943	0.347	−4.778	13.551
	Group 3	2.403	4.797	0.076	0.501	0.617	−7.047	11.852
	Group 4	1.292	4.662	0.041	0.277	0.782	−7.891	10.475
SNP4 rs1042522 ^†^	(Constant)	47.168	3.603		13.091	0.000	40.071	54.266
	Group 1	−3.174	2.164	−0.176	−1.467	0.144	−7.438	1.089
	Group 2	2.257	2.113	0.137	1.068	0.287	−1.905	6.420
	Group 3	0.896	2.132	0.051	0.420	0.675	−3.303	5.095
	Group 4	0.815	2.218	0.043	0.367	0.714	−3.554	5.184
SNP5 rs803580 ^††^	(Constant)	49.096	2.918		16.827	0.000	43.349	54.844
	Group 1	−4.981	1.951	−0.259	−2.553	0.011	−8.825	−1.138
	Group 2	1.254	1.860	0.072	0.674	0.501	−2.410	4.918
	Group 3	0.113	2.049	0.006	0.055	0.956	−3.924	4.150
	Group 4	−0.317	1.968	−0.016	−0.161	0.872	−4.195	3.560
SNP6 rs1799983 ^†††^	(Constant)	49.603	2.573		19.278	0.000	44.535	54.672
	Group 1	−6.357	2.258	−0.261	−2.815	0.005	−10.806	−1.909
	Group 2	1.151	2.104	0.053	0.547	0.585	−2.993	5.294
	Group 3	−0.717	2.264	−0.030	−0.317	0.752	−5.177	3.744
	Group 4	−0.681	2.098	−0.030	−0.325	0.746	−4.813	3.451

* Multiple linear regression: R = 0.257 (adjust R^2^ = 0.051), df (4), *p* = 0.002. Reference: Allele (AA). ** Multiple linear regression: R = 0.283 (adjust R^2^ = 0.065), df (4), *p* < 0.001. Reference: Allele (TT). *** Multiple linear regression: R = 0.278 (adjust R^2^ = 0.062), df (4), *p* = 0.001. Reference: Allele (TT). ^†^ Multiple linear regression: R = 0.264 (adjust R^2^ = 0.054), df (4), *p* = 0.002. Reference: Allele (GG). ^††^ Multiple linear regression: R = 0.290 (adjust R^2^ = 0.069), df (4), *p* < 0.001. Reference: Allele (TT). ^†††^ Multiple linear regression: R = 0.274 (adjust R^2^ = 0.060), df (4), *p* = 0.001. Reference: Allele (TT).

**Table 3 ijms-26-08447-t003:** Statin type and group interaction effect on ED.

Variants	Unstandardized Coefficients		Standardized Coefficients	t	Sig.	95.0% CI for B	
	B	Std. Error	Beta			Lower Bound	Upper Bound
(Constant)	49.460	1.705		29.015	0.000	46.102	52.818
Atorva_Group1	−10.566	2.779	−0.274	−3.802	0.000	−16.040	−5.091
Atorva_Group2	1.659	2.695	0.045	0.615	0.539	−3.651	6.968
Atorva_Group3	−0.808	2.624	−0.023	−0.308	0.758	−5.977	4.361
Simva_Group1	−4.723	3.541	−0.090	−1.334	0.184	−11.700	2.253
Simva_Group2	0.078	3.154	0.002	0.025	0.980	−6.135	6.291
Simva_Group3	3.903	4.421	0.058	0.883	0.378	−4.807	12.613

Multiple linear regression: R = 0.299 (adjust R^2^ = 0.063), df (6), *p* = 0.002. Reference: Control group.

**Table 4 ijms-26-08447-t004:** The effects of gene–drug interaction between statin type and genetic variants on ED.

Variants	Statin and Group	Unstandardized Coefficients		Standardized Coefficients	t	Sig.	95.0% CI for B	
		B	Std. Error	Beta			Lower Bound	Upper Bound
SNP1 rs4919686 *	(Constant)	49.134	1.544		31.827	0.000	46.093	52.175
	Atorva_Group1	−8.585	2.397	−0.249	−3.581	0.000	−13.308	−3.862
	Atorva_Group2	1.732	2.205	0.055	0.785	0.433	−2.613	6.076
	Atorva_Group3	−0.637	2.167	−0.021	−0.294	0.769	−4.906	3.633
	Simva_Group1	−3.353	3.013	−0.074	−1.113	0.267	−9.288	2.582
	Simva_Group2	−0.158	2.520	−0.004	−0.063	0.950	−5.123	4.806
	Simva_Group3	3.250	3.503	0.060	0.928	0.354	−3.650	10.151
SNP2 rs17703883 **	(Constant)	49.113	1.678		29.276	0.000	45.809	52.418
	Atorva_Group1	−4.611	1.275	−0.258	−3.617	0.000	−7.123	−2.100
	Atorva_Group2	1.003	1.340	0.054	0.749	0.455	−1.636	3.642
	Atorva_Group3	−0.231	1.304	−0.013	−0.177	0.860	−2.799	2.338
	Simva_Group1	−2.005	1.708	−0.079	−1.174	0.241	−5.369	1.359
	Simva_Group2	0.213	1.571	0.009	0.135	0.892	−2.881	3.307
	Simva_Group3	2.125	2.207	0.063	0.963	0.337	−2.222	6.473
SNP3 rs57989773 ***	(Constant)	48.742	1.631		29.893	0.000	45.530	51.954
	Atorva_Group1	−8.084	2.358	−0.242	−3.428	0.001	−12.729	−3.438
	Atorva_Group2	2.132	2.464	0.062	0.865	0.388	−2.723	6.987
	Atorva_Group3	0.227	2.491	0.007	0.091	0.928	−4.681	5.134
	Simva_Group1	−4.005	3.515	−0.077	−1.139	0.256	−10.930	2.919
	Simva_Group2	0.796	3.122	0.018	0.255	0.799	−5.354	6.946
	Simva_Group3	4.621	4.406	0.068	1.049	0.295	−4.058	13.300
SNP4 rs1042522 ^†^	(Constant)	49.072	1.583		31.007	0.000	45.955	52.190
	Atorva_Group1	−5.271	1.421	−0.260	−3.710	0.000	−8.070	−2.472
	Atorva_Group2	0.831	1.434	0.041	0.579	0.563	−1.994	3.656
	Atorva_Group3	−0.305	1.381	−0.016	−0.221	0.826	−3.026	2.416
	Simva_Group1	−1.349	2.093	−0.043	−0.645	0.520	−5.472	2.773
	Simva_Group2	0.334	1.720	0.013	0.194	0.846	−3.054	3.722
	Simva_Group3	2.120	2.355	0.058	0.900	0.369	−2.520	6.759
SNP5 rs803580 ^††^	(Constant)	49.467	1.523		32.486	0.000	46.467	52.466
	Atorva_Group1	−6.163	1.568	−0.270	−3.930	0.000	−9.253	−3.074
	Atorva_Group2	0.794	1.397	0.039	0.569	0.570	−1.957	3.546
	Atorva_Group3	−0.376	1.556	−0.017	−0.241	0.809	−3.442	2.690
	Simva_Group1	−3.395	1.963	−0.114	−1.730	0.085	−7.263	0.472
	Simva_Group2	0.120	1.909	0.004	0.063	0.950	−3.641	3.880
	Simva_Group3	1.581	2.538	0.040	0.623	0.534	−3.418	6.580
SNP6 rs1799983 ^†††^	(Constant)	49.611	1.483		33.450	0.000	46.689	52.532
	Atorva_Group1	−7.858	1.904	−0.281	−4.127	0.000	−11.609	−4.107
	Atorva_Group2	1.231	1.844	0.046	0.668	0.505	−2.402	4.864
	Atorva_Group3	−1.131	1.938	−0.040	−0.584	0.560	−4.949	2.686
	Simva_Group1	−2.937	2.667	−0.072	−1.101	0.272	−8.192	2.318
	Simva_Group2	−0.775	2.163	−0.024	−0.358	0.720	−5.036	3.485
	Simva_Group3	1.263	2.658	0.030	0.475	0.635	−3.974	6.500

* Multiple linear regression: R = 0.277 (adjust R^2^ = 0.053), df (6), *p* = 0.004. Reference: Control group and allele (AA). ** Multiple linear regression: R = 0.290 (adjust R^2^ = 0.061), df (6), *p* = 0.002. Reference: Control group and allele (TT). *** Multiple linear regression: R = 0.282 (adjust R^2^ = 0.057), df (6), *p* = 0.003. Reference: Control group and allele (TT). ^†^ Multiple linear regression: R = 0.281 (adjust R^2^ = 0.056), df (6), *p* = 0.003. Reference: Control group and allele (GG). ^††^ Multiple linear regression: R = 0.297 (adjust R^2^ = 0.065), df (6), *p* = 0.001. Reference: Control group and allele (TT). ^†††^ Multiple linear regression: R = 0.293 (adjust R^2^ = 0.063), df (6), *p* = 0.001. Reference: Control group and allele (TT).

**Table 5 ijms-26-08447-t005:** SNP incubation time and temperature.

SNP	Re Restriction Enzyme	Incubation Temperature (°C)	Incubation Time (min)
Rs1799983	Ban II	37	60
Rs1042522	BstUI	60	15
Rs8023580	HpyCH41V	37	15

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
