# Peer review of "Genetic Prediction of Erectile Dysfunction Caused by Statins Among Malaysian Cardiac Patients"

_ijms, 2025, doi:10.3390/ijms26178447_

Round 1
Reviewer 1 Report
Comments and Suggestions for Authors
This manuscript addresses an important clinical question regarding the potential influence of genetic polymorphisms on the development of erectile dysfunction (ED) among statin users, particularly in a Malaysian cardiac patient population. While the topic is relevant and the exploration of gene–drug interactions is of growing interest in personalized medicine, the study suffers from multiple conceptual, methodological, and analytical weaknesses. Each section of the manuscript requires substantial revision to meet the scientific and editorial standards expected for publication in a high-impact journal.
- The introduction is superficial and poorly structured. It lacks a logical flow from background to justification.
- The prevalence of ED in Malaysia is cited (81.5%) based on self-reported data, but no critique is offered about its potential overestimation or lack of clinical confirmation.
- There is redundancy: definitions and classifications of ED (types and severity) are repeated without advancing the scientific rationale.
- The discussion about statins is oversimplified. Contradictory findings from the literature are listed without critical analysis or mechanistic insight (e.g., endothelial dysfunction, testosterone suppression).
- The justification for a genetic study is weak. The selected genes and their biological plausibility are not introduced here, leaving the reader without a clear hypothesis.
- The aim is stated vaguely ("to investigate potential interactions...") and does not specify the genetic variants or expected direction of effect.
- The presentation of results is dense, tabular, and hard to interpret without figures or summary graphics (e.g., forest plots, genotype-specific IIEF score comparisons).
- Although six SNPs were tested across four patient groups, no correction for multiple comparisons was applied (e.g., Bonferroni or FDR), which is a major flaw given the number of subgroup analyses.
- The effect sizes are modest (e.g., ~5–10 points on IIEF), and the adjusted R² values are consistently low (mostly <0.07), suggesting weak explanatory power.
- The distinction between statistical vs. clinical significance is not addressed. A few point reductions in IIEF may not translate into meaningful changes in clinical practice.
- Grouping choices are questionable: for example, combining diabetics with statin users into a specific group without reporting HbA1c or glycemic control data weakens interpretability.
- Key confounders like age, testosterone levels, comorbidities, and medication use (e.g., beta-blockers) are not controlled for in the regression models.
- The discussion overstates the importance of modest genetic associations and fails to critically assess the limitations of the findings.
- Mechanistic pathways are briefly mentioned but not linked back to the specific SNPs studied (e.g., how exactly does NR2F2-AS1 influence ED under statin exposure?).
- The impact of atorvastatin on ED is highlighted, but the discussion does not explain why atorvastatin would have a different effect than simvastatin from a pharmacological or mechanistic perspective.
- The authors fail to acknowledge major limitations, such as:
- Cross-sectional design (no causal inference)
- Absence of hormonal or biochemical measures (e.g., testosterone, NO)
- No replication cohort
- Small sample size for subgroup analyses
- Lack of population stratification control (especially in a homogenous Malay cohort)
- The discussion includes irrelevant concepts, such as sperm motility and fertility, which are not part of the study outcomes.
- The conclusion claims the study presents “novel outcomes,” but novelty is questionable given the modest associations and lack of mechanistic insight.
- Statements such as "atorvastatin showed a significant negative effect..." are overgeneralized, considering the effect was not consistent across all SNPs and groups.
- The conclusion does not acknowledge the study’s exploratory nature and the need for replication in larger, well-controlled cohorts.
- It misses the opportunity to propose future directions (e.g., longitudinal studies, functional assays, pharmacogenomic screening strategies).
Reviewer 2 Report
Comments and Suggestions for Authors
This study aims to identify the genetic determinants of statin-associated erectile dysfunction (ED), which is poorly understood but clinically relevant considering the wide use of statins. Directing attention to a Malaysian population provides significant population-specific genetic findings and enriches the global diversity in pharmacogenomic investigations. The manuscript is clear about its purpose: to examine the relationship of genetic variants with the risk of ED in statin users, especially in patients treated with atorvastatin and simvastatin. The inclusion of a diagnosis of ED confirmed by a physician and IIEF scores provides greater clinical validation. With this stratification into four independent patient groups (with and without ED, with and without statin or diabetes), one can make a valuable stratified analysis. The addition of six genes (e.g., NR2F2-AS1, NOS3, CYP17A1) and several SNPs favors the hypothesis of a broad genetic range.
Below are some suggestions for improvement:
- The manuscript has multiple language errors.
- Several points are reiterated in the abstract and results sections (e.g., simvastatin had no effect) unnecessarily.
- There are some redundant columns in several tables (in particular Tables 2 and 4), and it is possible to compress them for better presentation.
- Baseline characteristics such as blood pressure, cholesterol levels, or testosterone are not included.
- The mechanistic basis of the selection of each gene/SNP can be stated more clearly in the introduction. For example, how were SNPs such as rs803580 selected in this context?
- Some of the references are obsolete.
Round 2
Reviewer 1 Report
Comments and Suggestions for Authors
The hypothesis could be more explicitly stated (e.g., specifying expected gene–drug interactions)
Some recent studies on pharmacogenomics and ED (e.g., Su et al. 2024, Front Endocrinol) should be discussed more thoroughly to situate the study within current research.
Given the predominance of Malay participants, the findings may not be generalizable to other ethnic groups. Replication in more diverse cohorts is warranted.
Key variables such as age, BMI, comorbidities, and concomitant medications were not fully adjusted in the regression models.
No adjustment was made for multiple comparisons despite the number of SNPs analyzed, raising the possibility of false-positive associations. Applying methods such as Bonferroni correction would strengthen the robustness of the results.
The current wording of the conclusions is too definitive given the study’s observational, cross-sectional design. Statements such as “atorvastatin showed a substantial adverse effect” should be reframed more cautiously to reflect that the findings are associative and hypothesis-generating rather than causal.
The associations identified for NOS3 and NR2F2-AS1, while statistically significant, have modest effect sizes (adjusted R² < 0.07), which raises concerns about their clinical relevance. This should be explicitly acknowledged in the discussion, with an emphasis on the need for replication in larger cohorts
Author Response
Reviewer 1
Comment 1: The hypothesis could be more explicitly stated (e.g., specifying expected gene–drug interactions)
Thanks a lot for this comment. This hypothesis for the effect of statins on the ED is added to the manuscript.
Page 2, line 68-70:
‘The hypothesis of the adverse reaction of statins on the ED can also be justified within the gene-drug interactions. The present study investigated several genes mentioned earlier that were hypothesised to interact with statins.’
Comment 2: Some recent studies on pharmacogenomics and ED (e.g., Su et al. 2024, Front Endocrinol) should be discussed more thoroughly to situate the study within current research.
Thanks a lot for this comment. Sentences are added to the discussion.
Page 9, line 230-238:
‘With regards to pharmacogenetic predictors, Su et al. identified the genetic basis of the lipid-lowering agent, which was significantly associated with ED [32]. They also found that the variants of the genes influenced ED and associated illnesses (like prostate hy-perplasia, prostate cancer, infertility, and others). They reported the causality, but there is a need for more information about the mechanism of action between statins and the genes that induce ED. Inversely, our study selected the genes most related to ED due to cholesterol synthesis, sex hormone production, and gonad development. Furthermore this study determined the associations between statins and gene variants of the ED within drug-gene interactions.’
Comment 3: Given the predominance of Malay participants, the findings may not be generalizable to other ethnic groups. Replication in more diverse cohorts is warranted.
Thanks a lot for this critical comment. A paragraph about the limitations of ethnicity is added.
Page 9, line 258-261:
‘Most of the participants in the study are Malays; therefore, the influence of ethnicity is a significant generalization limitation. The authors of the study recommended involving more ethnicities in a larger cohort to determine the differences in the incidence and genetic associations with the ED.’
Comment 4: Key variables such as age, BMI, comorbidities, and concomitant medications were not fully adjusted in the regression models.
Thanks a lot for this comment. These variables were analysed in univariate analyses but there was no significant effect of the demographic characteristics and concomitant diseases on the ED. The impact of drug-drug interactions on the ED also showed no significance. Therefore these variables were not adjusted in the regression models.
Comment 5: No adjustment was made for multiple comparisons despite the number of SNPs analyzed, raising the possibility of false-positive associations. Applying methods such as Bonferroni correction would strengthen the robustness of the results.
Thanks a lot for this excellent comment. The linear regression showed the most authentic statistical test. The Bonferroni correction could also be used in linear regression. Through statistical checking, the values of the results have remained within standard values, indicating that the significant outcomes of the present study are significant without any type error. The Bonferroni correction can be directly checked by multiplying each p-value by the number of tests or by dividing the adjusted alpha by the number of tests. The first method is used, and all outcomes obtained are less than 0.05.
Comment 6: The current wording of the conclusions is too definitive given the study’s observational, cross-sectional design. Statements such as “atorvastatin showed a substantial adverse effect” should be reframed more cautiously to reflect that the findings are associative and hypothesis-generating rather than causal.
Thanks a lot for this comment. The sentence is corrected.
Page 12, line 358-359:
‘Atorvastatin showed potential risk of adverse effect related to erectile function as compared to simvastatin,..’
Comment 7: The associations identified for NOS3 and NR2F2-AS1, while statistically significant, have modest effect sizes (adjusted R² < 0.07), which raises concerns about their clinical relevance. This should be explicitly acknowledged in the discussion, with an emphasis on the need for replication in larger cohorts
Thanks a lot for this comment. Sentences are added in the manuscript.
Page 8, line 217-220:
‘The association found in this study, however, is justified by the small sample size of patients with gene variants. There is a need for studies to investigate the effect of these genes with atorvastatin in a prospective design to get more robust associations and mechanisms on the ED.’
Page 9, line 232-233:
‘This result, however, need to be interpreted in caution due to modest effect sizes, which may need to be replicated in larger cohorts.’
Reviewer 2 Report
Comments and Suggestions for Authors
The authors have satisfactorily revised the manuscript and addressed the reviewers' comments. I recommend it for acceptance
Author Response
Thank you for the approval of the revised manuscript.
Round 3
Reviewer 1 Report
Comments and Suggestions for Authors
I have no comments